# The Effect of Entrepreneurship on Start-Up Open Innovation: Innovative Behavior of University Students

**Jongwan Lee** [1]**, Daesu Kim** [1] **and Sanghyun Sung** [2,*]

[1]   School of Business, Yeungnam University, Gyeongsan 38541, Korea; jwlee2@ynu.ac.kr (J.L.); kdsduck@ynu.ac.kr (D.K.)

[2]   POSTECH Entrepreneurship Center, Pohang University of Science and Technology, Pohang 37673, Korea

[*]   Correspondence: flyparis@postech.ac.kr; Tel.: +82-54-279-8451

**Abstract:** This study investigates whether higher start-up intention leads to innovative behavior and whether innovative behavior is increased by the medium of entrepreneurship. This study tested the hypotheses by conducting a survey on students participating in entrepreneurship club activities in university. The results showed that start-up intention affects innovative behavior and has a significant effect on the sub-factors of entrepreneurship, such as innovation, risk taking, and proactiveness. The result of an analysis of the mediating effect of entrepreneurship on innovative behavior showed that all sub-factors performed a partially mediating role. It can therefore be said that higher start-up intention leads to more innovative behavior and that entrepreneurship serves as an important link in this relationship. These results show that increasing start-up intention may lead to innovative behavior and imply that this has educational relevance in cultivating entrepreneurship. However, this study is limited in terms of the generalizability of the results, as the subjects are university students participating in entrepreneurship club activities in Korea. Therefore, more significant outcomes can be obtained in further research by targeting a broader scope of subjects.

**Keywords:** start-up intention; entrepreneurship; innovative behavior

---

## 1. Introduction

### 1.1. Background

In current times, new entrants into the labor market prefer job security and are losing diversity in career choice. It has therefore become critical to increase opportunities to create new value through start-up companies and to promote flexibility in the labor market. It is important to facilitate a start-up ecosystem that can create new values through start-up assistance and promotion policies for the rising generation—who prefer jobs such as public servants or working in large corporations—to make career choices. There are various possible approaches to securing diversity in employment. However, the most important aspect is to change the typical forms of employment by structurally improving the conventional labor market. In other words, it is necessary to individualize the collectivized job paradigm and diversify employment opportunities. In this sense, it is essential to turn the current system of collective and standardized employment opportunities into an individualized and customized system of employment through business start-up.

From this perspective, creating the desired start-up culture to build an appropriate start-up ecosystem not only enables the ecosystem to be structured in a virtuous cycle but also achieves sustainable development of the national economy and increases diversity in the labor market. A virtuous

cycle accelerates positive competition among both newly established firms and established ones while simultaneously creating a positive environment that increases innovation [1]. Before promoting start-up, it is necessary to increase the desire to start a new business that can enhance competencies. Start-up intention represents this potential for future start-up and is aimed at fulfilling the ideals and goals one pursues.

Entrepreneurship not only seeks effective innovation but also increases competitiveness and productivity [2]. There is a significant relationship between entrepreneurial motivation to undertake entrepreneurship and innovative behavior, and it is both intrinsic and extrinsic motivation that affects the innovative behavior [3]. This is because people with high entrepreneurship have great internal control, are innovative, and tend to be highly forward thinking [4]. People with a particularly defiant and risk-taking attitude promote innovative activities by more boldly investing time and effort into these activities [5]. In particular, people with high entrepreneurship have a positive problem-solving ability in their innovation behavior [6]. Above all, innovation is considered to be crucial because it serves as an important way of securing a competitive advantage and producing excellent results [7]. It also contributes new products by combining new knowledge with existing knowledge as well as opportunities and insights to open new markets. Lastly, innovation also encourages people to understand new technological trends and available opportunities [8].

In recent times it has been proved that the invigoration of entrepreneurship is an important factor that affects behavioral outcomes through the perception of members, and more diverse approaches are required regarding the relationship between entrepreneurship and innovative behavior [9]. Amabile and Conti proved the effect of entrepreneurship on innovation through empirical research [10].

Existing studies on start-up intention mostly address the effects of endogenous factors—such as self-efficacy, disposition, and behavior—on start-up intention, or are conducted within an exogenous scope such as the perception of start-up or the social environment. However, there has to date not been significant research on the actual effects of start-up intention. While most studies aim to identify antecedent factors, this study examines the concept of expansion in start-up intention and to what extent it expands.

This study investigates whether higher start-up intention leads to innovative behavior and whether innovative behavior is increased by the medium of entrepreneurship. This study also tests that start-up intention affects innovative behavior and has a significant effect on the sub-factors of entrepreneurship.

### 1.2. Theoretical Discussions and Hypothesis Setting

#### 1.2.1. Start-Up Intention and Entrepreneurship

In general, human beings undergo a social cognitive process as they perceive possible alternatives, mostly when they encounter a new environment that they need to adapt to. People have different ways of processing information, depending on their cognitive style [11]. People with more flexible cognitive abilities have higher adaptability to a new situation, and they tend to adapt while showing the will to effectively explore and select various methods [12]. In the context of start-up intention, this cognitive flexibility enables people to show diverse behaviors to survey a problem from various perspectives, to solve a problem creatively, and to define the existing problem in a new way to find the best solution. This type of person is more open to change, has a high preference for new things, has little fear of new experiences, and tries to find more innovative solutions without restrictions when facing a problem and in the process of solving it [13].

According to the theory of planned behavior, intention serves as a powerful antecedent for behavior and appears in the form of materialized behavior [14]. In Ajzen's theory of planned behavior, intention is a product of interaction among attitudes toward specific behavior, subjective norms, and perceived control [15]. According to this theory, intention is closely related to entrepreneurial behavior because the entrepreneurial process is planned [16].

Entrepreneurship is also extremely important in terms of the effort that is shown in business start-up or expansion. This positively links entrepreneurship with start-up intention [17]. However, in terms of decision making and implementation, entrepreneurship can be regarded as a competency that is necessary after start-up because people with the will to start a business are likely to have inherent entrepreneurship or similar characteristics [18]. This leads to our first hypothesis:

**Hypothesis 1.** *Start-up intention has a positive effect on entrepreneurship.*

### 1.2.2. Start-Up Intention and Innovative Behavior

In his work, Kanter claimed that innovative behavior starts from being aware of a problem, choosing a new idea, and solving a problem [19]. In business studies, innovative behavior refers to activities related to creating, adopting, or applying new ideas to improve work and performance [20]. Innovative behavior consists of three steps: (1) seeking new ideas to perceive and solve a problem, (2) checking legitimacy to specifically implement the ideas discovered in the previous step and discovering ways to support them inside or outside of the organization, and (3) devising an innovative model fit for the specific organizational characteristics to be applied to an individual's job or related department or even the entire organization, thereby finding solutions to the problem [21]. This innovative behavior starts at the process of seeking new and useful ideas in all areas [22]. New ideas revealed in this process emerge from adopting, combining, and applying knowledge that already exists [23].

People who display innovative behavior have opportunities to explore innovation-related knowledge and use it to gain new insights and to develop their competencies. During this process, they also reveal innovative ideas using available knowledge [24]. Furthermore, the effective use of existing knowledge as well as the integration of new knowledge can further promote innovative behavior [25].

Many previous studies on the factors that induce innovative behavior showed that individuals, relationships, tasks, and organizational characteristics have all kinds of effects. Studies on individual factors have emphasized that important factors include not only the cognitive style or openness of individuals but also a creative personality, self-efficacy, job satisfaction, and continuous learning activity [26].

Regarding the fact that intention comes before behavior, Ajzen and Fishbein stated that will is the most apparent predictor that leads to behavior after empirically analyzing the correlation through the theory of planned behavior [27]. In other words, start-up intention is the factor that best explains the behaviors related to starting a business.

Some studies have argued that the relationship between start-up intention and innovation is not considered to be significant [28]. From this perspective, it is necessary to verify once again the effects of start-up intention on the fulfillment of ideals and goals in the previous step where innovative behavior was addressed. Krueger, Reilly, and Carsrud claimed that intention is the strongest explanatory variable of behavior [29]. Accordingly, the following hypothesis is established based on the results of the literature review:

**Hypothesis 2.** *Start-up intention has a positive effect on innovative behavior.*

### 1.2.3. Entrepreneurship and Innovative Behavior

Entrepreneurship is an organized process of taking a risk while focusing on innovation without losing customers or market opportunities to become more competitive [30]. Entrepreneurship generally does not appear out of the blue but rather during the process of solving problems that arise in one or more events under specific circumstances.

Entrepreneurship is a type of mental, physical, and situational experience manifested in the process of turning a series of uncertain situations into certain ones. From a mental point of view,

entrepreneurship is a substance of individual belief and intuitive expression that is not produced naturally but happens due to mental and situational experiences. Innovation is an outcome of this substance. Nadkarni described innovative activities as the process of entrepreneurship, and Brenkert claimed that innovation originates from entrepreneurship [31,32]. In other words, entrepreneurship is a part of the innovation process and, in a broader scope, a critical factor that determines economic performance [33].

Brokel and Binder argued that all innovative behaviors are attempts at new opportunities or activities and that these behaviors are accompanied by knowledge inquiry and use [34]. According to Person, entrepreneurship aims for effective innovation while also increasing competitiveness and productivity [2]. This is because people with entrepreneurship skills have great internal control, are innovative, and tend to be highly forward thinking [3]. Amabile and Conti proved the effect of entrepreneurship on innovation through their empirical study, and Song claimed that people with defiant, active, and risk-taking attitudes promote innovative activities by making bold investments [5,10].

Innovation is considered to be the most important source not only for a competitive advantage but also for producing excellent results [6]. This is because combining new knowledge with existing knowledge provides opportunities and insight to develop new products and expand the market. Moreover, it is also because members can understand the trends of new technological changes by using this knowledge and conducting innovative behavior by taking more advantage of these opportunities [8]. This leads us to our third hypothesis:

**Hypothesis 3.** *Entrepreneurship plays a mediating role between start-up intention and innovative behavior.*

*1.3. Research Method and Results*

We used statistics program SPSS Version 23 for hypothesis testing of this research model. Most students who start a business in university participate in an entrepreneurship club. Therefore, in order to verify the suggested research model, the researcher conducted a survey on university students participating in entrepreneurship club activities in Korea.

## 2. Materials and Methods

*2.1. Sample and Data Collection*

To test our hypothesis, we conducted a survey on students of entrepreneurship clubs at two universities in Gyeongsangbuk-do. In particular, we surveyed two representative universities with the largest start-ups in the region and analyzed the data collected in the field by handing out questionnaires (See Appendix A) directly to university students conducting club activities during June 2017. Of the total of 182 copies of the questionnaire that were distributed, 171 copies were used in the analysis. We excluded 11 copies containing insincere responses. The respondents comprised 125 male students (73.1%), 44 female students (25.7%), and 2 others. There were 39 freshmen (22.8%), 32 sophomores (18.7%), 29 juniors (17.0%), 69 seniors (40.4%), and 2 others. Of the respondents, 73 students were majoring in humanities and social sciences (42.7%), 96 in natural sciences and engineering (56.2%), and 2 in other departments.

*2.2. Measurement of Variables*

2.2.1. Start-Up Intention

It is worth noting that in the start-up intention to start a business, all human behaviors (including business-related behaviors) take precedence and are related to the composition of the various elements, and there may be several external or internal motives [35]. It is an important task for researchers to search for factors influencing business intentions if they are one of the variables that trigger

business-related behavior [36]. In addition, most of the potential entrepreneurs (i.e., people who plan to carry out their business activities in the near future) are found among students. The important thing is that a positive attitude and experience of business start-ups should precede university students making business decisions and actions [37]. Start-up intention refers to the will to manage a business or the intention to start a company. In this study, start-up intention is measured by using the items provided by Kwon and Yoon [38].

### 2.2.2. Entrepreneurship

Entrepreneurship refers to an indomitable attitude and ability to overcome change and uncertainty with future orientation, capturing and pursuing new opportunities through innovation, and proactiveness and risk taking without being bound by limited resources.

This study modified and used the survey items developed by Covin and Slevin, measuring the three characteristics of entrepreneurship such as innovation, risk taking, and proactiveness with each item [39]. The operational definitions of the components of entrepreneurship are as follows:

1. Innovation. Innovation refers to the preference and pursuit of change and the reform of ideas or products to secure a competitive advantage. Innovation is measured according to the overall perception of research and development, the pursuit of innovation, acceptance of employees' original and innovative ideas, development of creative marketing methods for products or services, and free and active communication between higher and lower positions.

2. Risk taking. Risk taking refers to taking on a challenge and carrying the business forward despite the risk of failure due to low certainty. Risk taking is measured according to the overall perception of new business entry despite risks and uncertain circumstances, pursuit of growth over stability, and executing a relatively high-profit project despite high risks.

3. Proactiveness. Proactiveness indicates acting aggressively before other firms that are in competition. In other words, proactiveness is measured according to the overall perception of the pursuit of new product development, the revelation of business ideas, and being ahead of competitors in taking active measures.

### 2.2.3. Innovative Behavior

Innovative behavior refers to various specific individual behaviors—including the creation and implementation of ideas—that support and develop the ideas of other members of the organization while finding suitable ideas from existing technologies and methods or creating completely new ideas [21].

## 3. Results

### 3.1. Validity and Reliability of Variables

For the empirical analysis of the research model, a principal component analysis was conducted to determine how reliable and valid the constructs of each variable are. Moreover, a varimax rotation was used for the factor analysis of the constructs, simplifying the loadings of each factor and securing independence. The internal consistency and reliability of the items was verified using Cronbach's alpha. This study applied 0.6 or higher as the standard for acceptable reliability.

As a result of the factor analysis, there were eight items for start-up intention (independent variable) with a Cronbach's α of 0.946. Concerning entrepreneurship (mediator variable), there were four items for innovation with a Cronbach's α of 0.865, three items for risk taking with 0.779, and three items for proactiveness with 0.639 as Cronbach's α. Accordingly, the items of entrepreneurship as the mediator variable showed high reliability. Finally, there were eight items for innovative behavior (dependent variable) with Cronbach's α of 0.918, showing high reliability. As presented in Table 1, the Cronbach's α values of all variables are higher than 0.639, showing that the items have high reliability.

**Table 1.** Validity and reliability analysis of variables.

|  | Factor 1 | Factor 2 | Factor 3 | Factor 4 | Factor 5 |
|---|---|---|---|---|---|
| Start-up Intention 3 | **0.836** | 0.219 | 0.101 | 0.177 | 0.047 |
| Start-up Intention 5 | **0.830** | 0.172 | 0.139 | 0.085 | 0.059 |
| Start-up Intention 4 | **0.823** | 0.181 | 0.234 | 0.130 | 0.172 |
| Start-up Intention 6 | **0.794** | 0.161 | 0.141 | 0.205 | 0.061 |
| Start-up Intention 2 | **0.788** | 0.180 | 0.256 | 0.171 | 0.166 |
| Start-up Intention 1 | **0.778** | 0.224 | 0.250 | 0.189 | 0.128 |
| Start-up Intention 8 | **0.694** | 0.335 | 0.124 | 0.282 | −0.084 |
| Start-up Intention 7 | **0.672** | 0.296 | 0.225 | 0.224 | 0.023 |
| Innovative Behavior 2 | 0.107 | **0.763** | 0.312 | 0.133 | 0.164 |
| Innovative Behavior 8 | 0.262 | **0.735** | 0.101 | 0.140 | 0.132 |
| Innovative Behavior 7 | 0.270 | **0.728** | 0.156 | 0.150 | 0.093 |
| Innovative Behavior 4 | 0.205 | **0.727** | 0.187 | 0.207 | 0.077 |
| Innovative Behavior 3 | 0.107 | **0.726** | 0.361 | 0.006 | 0.099 |
| Innovative Behavior 9 | 0.421 | **0.721** | −0.032 | 0.059 | 0.070 |
| Innovative Behavior 1 | 0.134 | **0.683** | 0.385 | 0.206 | −0.016 |
| Innovative Behavior 5 | 0.267 | **0.624** | 0.106 | 0.402 | 0.122 |
| Entrepreneurship 2 | 0.226 | 0.319 | **0.789** | 0.036 | 0.040 |
| Entrepreneurship 1 | 0.292 | 0.309 | **0.752** | 0.076 | 0.135 |
| Entrepreneurship 3 | 0.317 | 0.382 | **0.666** | 0.209 | 0.042 |
| Entrepreneurship 4 | 0.298 | 0.131 | **0.624** | 0.367 | 0.185 |
| Entrepreneurship 5 | 0.261 | 0.268 | 0.095 | **0.753** | 0.151 |
| Entrepreneurship 6 | 0.310 | 0.198 | 0.202 | **0.744** | 0.109 |
| Entrepreneurship 8 | 0.368 | 0.215 | 0.116 | **0.550** | 0.182 |
| Entrepreneurship 11 | −0.070 | 0.109 | −0.079 | 0.129 | **0.756** |
| Entrepreneurship 9 | 0.143 | 0.029 | 0.192 | 0.221 | **0.711** |
| Entrepreneurship 10 | 0.278 | 0.250 | 0.190 | −0.044 | **0.702** |
| Eigen Value | 6.057 | 5.096 | 2.919 | 2.279 | 1.872 |
| Variation | 23.295 | 19.599 | 11.227 | 8.766 | 7.201 |
| Cronbach's Alpha | 0.946 | 0.918 | 0.865 | 0.779 | 0.639 |

*3.2. Correlation of Variables*

For the analysis, we verified the correlation among the variables. The results are shown in Table 2. First, in the relationship between start-up intention (independent variable) and the sub-factors of entrepreneurship (mediator variable), there were significant positive correlations between start-up intention and the sub-factors of entrepreneurship, such as innovation with $r = 0.617$ ($p < 0.01$), risk taking with $r = 0.644$ ($p < 0.01$), and proactiveness with $r = 0.315$ ($p < 0.01$). Next, start-up intention had a significant positive correlation with innovative behavior (outcome variable) with $r = 0.590$ ($p < 0.01$). Finally, there was a significant positive correlation between innovative behavior (outcome variable) and the sub-factors of entrepreneurship (mediator variable) such as innovation with $r = 0.649$ ($p < 0.01$), risk taking with $r = 0.608$ ($p < 0.01$), and proactiveness with $r = 0.341$ ($p < 0.01$).

**Table 2.** Correlation analysis of variables.

| | M | SD | 1 | 2 | 3 | 4 | 5 |
|---|---|---|---|---|---|---|---|
| 1. Start-up Intention | 3.53 | 0.876 | 1 | | | | |
| 2. Innovation | 3.37 | 0.778 | 0.617 ** | 1 | | | |
| 3. Risk Taking | 3.56 | 0.729 | 0.644 ** | 0.566 ** | 1 | | |
| 4. Proactiveness | 3.33 | 0.709 | 0.315 ** | 0.343 ** | 0.348 ** | 1 | |
| 5. Innovative Behavior | 3.50 | 0.638 | 0.590 ** | 0.649 ** | 0.608 ** | 0.341 ** | 1 |

$* p < 0.05; ** p < 0.01$.

### 3.3. Testing of Research Hypotheses

The results of the analysis on the mediating effect of entrepreneurship show that all effects of start-up intention (step 1) on entrepreneurship were significant (Table 3). This means that—considering the effects of start-up intention on entrepreneurship—the standardized regression coefficient of innovation was significant at 0.603 with the $t$-value at 9.545 ($p < 0.001$), the standardized regression coefficient of risk taking was significant at 0.612 with the $t$-value at 9.914 ($p < 0.001$), and the standardized regression coefficient of proactiveness was significant at 0.288 with the $t$-value at 3.735 ($p < 0.001$). Therefore, Hypothesis 1 was accepted, and step 1 of the testing of the mediating effect was fulfilled, thereby verifying the relationship between independent and dependent variables in step 2. The result showed that—in the effects of start-up intention on innovative behavior—the standardized regression coefficient was significant at 0.590 with the $t$-value at 9.198 ($p < 0.001$), thereby satisfying the conditions for analysis in step 2. This means that Hypothesis 2 was accepted, and an analysis was conducted while controlling for the independent and mediator variables in the same time (step 3). The result showed that in the case of innovation for innovative behavior, the standardized regression coefficient of the independent variable in step 3 was significant at 0.319 with a $t$-value of 4.431 ($p < 0.001$), showing significance of the mediator variable and verifying the partial mediating effect of innovation. Next, considering risk taking, the standardized regression coefficient of the independent variable was significant at 0.341 with a $t$-value of 4.544 ($p < 0.001$), showing significance of the mediator variable and verifying the partial mediating effect of risk taking. Finally, for proactiveness, the standardized regression coefficient of the independent variable was significant at 0.527 with the $t$-value at 8.150 ($p < 0.001$), showing significance of the mediator variable and verifying the partial mediating effect of proactiveness. Therefore, the mediating effects described in Hypothesis 3 were accepted.

**Table 3.** Mediation regression analysis of entrepreneurship.

| Model | Innovation | | Risk Taking | | Proactiveness | | Innovative Behavior | | Innovative Behavior | | | | | |
|---|---|---|---|---|---|---|---|---|---|---|---|---|---|---|
| | β | t-Value | β | t-Value | β | t-Value | β | t-Value | β | t-Value | β | t-Value | β | t-Value |
| | | 6.368 *** | | 8.755 *** | | 10.478 *** | | 9.879 *** | | 7.026 *** | | 5.760 *** | | 5.697 *** |
| Gender | −0.080 | −1.243 | −0.113 | −1.802* | −0.42 | −0.542 | −0.030 | −0.459 | 0.006 | 0.101 | 0.016 | 0.267 | −0.021 | −0.327 |
| Major | 0.041 | 0.658 | −0.035 | −0.570 | −0.091 | −1.197 | 0.162 | 2.559 ** | 0.143 | 2.517 ** | 0.176 | 3.013 *** | 0.182 | 2.954 *** |
| Start-up Intention | 0.603 | 9.545 *** | 0.612 | 9.914 *** | 0.288 | 3.735 *** | 0.590 | 9.198 *** | 0.319 | 4.431 *** | 0.341 | 4.544 *** | 0.527 | 8.150 *** |
| Innovation | | | | | | | | | 0.449 | 6.275 *** | | | | |
| Risk Taking | | | | | | | | | | | 0.408 | 5.430 *** | | |
| Proactiveness | | | | | | | | | | | | | 0.218 | 3.455 *** |
| R | 0.629 | | 0.651 | | 0.317 | | 0.614 | | 0.706 | | 0.688 | | 0.648 | |
| $R^2$ | 0.395 | | 0.424 | | 0.100 | | 0.377 | | 0.499 | | 0.473 | | 0.420 | |
| Adj $R^2$ | 0.384 | | 0.413 | | 0.084 | | 0.366 | | 0.486 | | 0.460 | | 0.405 | |
| F | 35.509 *** | | 39.933 *** | | 6.052 *** | | 32.876 *** | | 40.305 *** | | 36.338 *** | | 29.297 *** | |
| Durbin–Watson | 2.283 | | 2.148 | | 2.060 | | 2.238 | | 2.190 | | 2.218 | | 2.131 | |

\* $p < 0.10$; \*\* $p < 0.05$; \*\*\* $p < 0.01$.

## 4. Discussion

This study examined the role of start-up intention and entrepreneurship in innovative behavior. The results showed that start-up intention and entrepreneurship both had significant effects on innovative behavior. Start-up intention had significant effects on the innovation, risk taking, and proactiveness of entrepreneurship and was positively correlated with innovative behavior. The results also show that entrepreneurship plays a partially mediating role in the relationship between start-up intention and innovative behavior, thereby proving the positive role of entrepreneurship.

According to the theory of planned behavior, start-up intention takes priority in the process of becoming an entrepreneur and has either a strong or a weak effect on behavior or practice, depending on the intention. The preference for new things and for solving problems along with openness to new experiences are expressed in the form of behavior. As shown in the results of this study, higher start-up intention not only leads to greater motivation for entrepreneurship but also encourages more innovative behavior.

One cannot start a business based only on a large amount of physical capital. Appropriate intellectual capital must rather be accumulated for start-up [40]. This must also be accompanied by the challenge of newly acquired knowledge. Start-up intention starts with a change of existing things and emergence of new phenomena. In this sense, innovation, proactiveness, and risk taking in entrepreneurs accelerate innovative behavior or help entrepreneurs to fulfill their intentions.

Entrepreneurship as capital is accumulated and manifested through individual and social acceptance as well as sharing, thereby producing both social and economic results [41–43]. While the approach to entrepreneurship has thus far been limited to the scope of enterprises and firms, in the future, a more universal approach must be taken. To this end, it is necessary to perceive and reveal entrepreneurship at the level of an entire organization to which individuals belong. In other words, there is a need for more a more in-depth and broad understanding of entrepreneurship by expanding its concept [44]. As a significant prerequisite for society, entrepreneurship must be displayed by all individuals [45]. Its dynamics must be increased by expanding the concept of entrepreneurship from only business to society in general. To this end, it is necessary to first understand the essence, attributes, and qualities of entrepreneurship. Since entrepreneurship affects both individual and community life, it must be considered as more significant, especially because it plays a crucial role in creating social

wealth [46,47]. Therefore, entrepreneurship should be understood and studied holistically, and people must be provided with education and training concerning entrepreneurship [48].

Education on entrepreneurship should be approached in terms of human development to enable people to think and make decisions flexibly, instead of taking a standardized and structuralized approach [49]. Strategic education on entrepreneurship must be provided to bring change in multiple aspects to various elements, such as an individual's competencies or knowledge as well as attitudes and perceptions [50]. Education on entrepreneurship at an earlier stage is more effective, because it causes a change in perception about individual and occupational success and helps people think they can change their desires and needs in life by creating values. Moreover, education on entrepreneurship plays a significant part in helping people clearly understand the difference between imagination and practice and display leadership for a self-directed life [48].

Innovative behavior is a process of materialization or concretization that acts by putting new ideas into practice. Intentions and spirit are powerful incentives in this process, driving people towards specific behaviors. To promote innovative behavior, it is first necessary to obtain different types of information and to secure flexibility in decision making, accompanied by education on innovation [19]. Innovation forces individuals to put new ideas into practice. Therefore, it is very important not only for individuals to make an effort but also for society or an organization to create the right atmosphere. Since innovation does not happen in an instant, an individual must have enough existing knowledge but also simultaneously explore enough new knowledge. By doing so, they conduct innovative behavior in related activities in addition to enhancing the ability to flexibly deal with situations [51,52].

Not everything is expressed as innovative through education on innovation. As there are individual differences in innovative behavior, it is necessary to reinforce intrinsic motivation. Intrinsic motivation encourages individuals to show enthusiasm for learning and to make the effort to think of new alternatives and ideas [53] because those with innovative behavior do their absolute best to pursue individual studies and to achieve perceived goals [54].

## 5. Conclusions

In conclusion, to increase the sustainability of firms through start-up and to grow a desirable start-up culture, it is necessary to bring change to the awareness or perception of people about start-up and to create a good-quality entrepreneurial environment. It is important to improve the function of cultivation and metacognition through education on entrepreneurship in that it is inseparable from the cultivation of entrepreneurship from a willingness to start a business [55,56]. After all, the success of a start-up is an effort for innovation [57,58].

By taking advantage of the fact that universities have appropriate educational infrastructures in multiple aspects, there is a need to increase individuals' motivation for start-up in university and to develop and utilize various programs to reinforce the network that expands entrepreneurial experience. Furthermore, it is important to increase the practicality of start-up by developing more concrete and systematic entrepreneurship programs.

These results show that increasing start-up intention may lead to innovative behavior and imply that this has educational relevance in cultivating entrepreneurship. However, this study is limited in terms of the generalizability of the results, as the subjects are university students participating in entrepreneurship club activities in Korea. Therefore, more significant outcomes can be obtained in further research by targeting a broader scope of subjects.

This study is aimed at university students who have intention to start a business in the actual idea stage of business start-up. Therefore, this study could be analyzed regardless of the respondents' major direction. In future research, as the target population is expanded to the general public, it is necessary to analyze the group by dividing the group based on the individual's field of study and practical ability. The scope of consideration of entrepreneurship variables is so wide that this study selected some of the most important entrepreneurship sub-factors. In future research, we will analyze the sub-factors of entrepreneurship variables and conduct analysis with entrepreneurship as one variables.

From the analysis methodology point of view, it will be possible to further segment the respondent group and perform advanced analysis using multi-group analysis. For example, it is possible to compare retirement groups with social beginner groups, or to analyze groups based on a sample of their major field, and derive meaningful results. In addition, the analysis is limited to Korea, but it can be compared with countries where industrialization is accelerating.

**Author Contributions:** J.L. and D.K. are from Yeungnam University, Republic of Korea. S.S. (corresponding author) is from Pohang University of Science and Technology (POSTECH), Republic of Korea. They designed the research conceptual model, wrote the literature review, collected the survey data, and interpreted the survey results together.

**Funding:** This research was supported by Basic Science Research Program through the National Research Foundation of Korea (NRF) funded by the Ministry of Education [Grants No. NRF-2019R1I1A1A01059142]. This work [Grants No.10054744] was supported by Business for University Entrepreneurship Center, funded Ministry of SMEs and Startups in 2016.

**Conflicts of Interest:** The authors have no conflict of interest.

## Appendix A.

### *Appendix A.1. Questionnaire*

The response scales use anchors such as 1 = Strongly Disagree, 2 = Disagree, 3 = Neutral, 4 = Agree, 5 = Strongly Agree.

**A.    Entrepreneurship**

1.    I constantly care about myself and think of creative ideas about new things.
2.    I'm creative and have an innovative way of thinking.
3.    I always try to find original and innovative ideas.
4.    I have no fear of new challenges.
5.    If I have to do it, I do it at any risk.
6.    Even in uncertain circumstances, the decision is made rather than hesitation.
7.    When I detect danger in my work, I actively find a way to overcome it.
8.    I think that you have to be aggressive at your own risk in order to explore potential opportunities.
9.    I like new things better than old ones and seek new trends.
10.    I think of it as progressive and reformative rather than conservative.
11.    I think it is unnecessary to be bound by tradition or old habits.

**B.    Innovation Behavior**

1.    I develop ideas to approach in new ways to solve difficult problems related to work (classes, learning activities, etc.).
2.    I try to find new ways to be used to perform my work (classes, learning activities, etc.).
3.    I consider creative solutions to work-related problems (classes, learning activities, etc.).
4.    I try to get support for innovative ideas in my work (classes, learning activities, etc.).
5.    I try to build empathy for innovative ideas in relation to work (classes, learning activities, etc.).
6.    I try to make key people in my school a sponsor for innovative ideas in terms of work (classes, learning activities, etc.).
7.    I refine my innovative ideas with regard to work (classes, learning activities, etc.) so that they can be useful.
8.    I try to introduce innovative ideas into work (classes, learning activities, etc.)  in a systematic way.

9.    I detail the practical value of innovative ideas in relation to work (classes, learning activities, etc.).

**C.    Start-Up Intention**

1.    I will challenge myself to be an entrepreneur in the future.
2.    I am excited when I think of starting a business.
3.    I have a lot of passion for starting a business.
4.    I plan to run my business one day.
5.    I want to be a manager by starting a business rather than being an employee.
6.    I will start a business even if there are a lot of risks.
7.    I would like to start a business whenever I have a great item.
8.    I think start-ups are attractive despite the risk of failure.

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
