# Peer review of "The Effect of Entrepreneurship on Start-Up Open Innovation: Innovative Behavior of University Students"

_2199-8531, doi:10.3390/joitmc5040103_

Round 1

Reviewer 1 Report

This author investigates whether higher start-up intention leads to innovative behavior and whether the effect is mediated by sub-factors of entrepreneurship, namely innovation, risk-taking, and proactiveness. By calculating a sample of 171 students participating in entrepreneurship club activities, the author gains the conclusion of entrepreneurship has a partial mediation effect on the relationship between start-up intention and innovative behavior. Technically it is a well-done study, but if I interpret one of the research results from Table 3 as “start-up intention affects the innovation behavior partially through innovation,” it may sound strange to common readers. Therefore I would suggest the author reconsider whether to select innovation as a mediating variable to examine, even though it is a sub-factor of entrepreneurship. Or should entrepreneurship be considered as one variable?

Author Response

Point 1: Technically it is a well-done study, but if I interpret one of the research results from Table 3 as “start-up intention affects the innovation behavior partially through innovation,” it may sound strange to common readers. Therefore I would suggest the author reconsider whether to select innovation as a mediating variable to examine, even though it is a sub-factor of entrepreneurship. Or should entrepreneurship be considered as one variable?

Response 1: Thank you for this comment. In this study, the mediating effect is analysed by comparing the regression coefficient values of the independent variables derived from steps 2 and 3. If the standardized regression coefficient of the step 2 is larger than the standardized regression coefficient of the step 3 independent variable, it is judged to have a mediating effect. In Table 3, since the independent variable is significant and lower than step 2, it can usually be interpreted as a partial mediating effect. The scope of consideration of entrepreneurship variables is so wide that this study selected some of the most important entrepreneurship sub-factors. In future research, we will analyse the sub-factors of entrepreneurship variables and conduct analysis with entrepreneurship as one variables. Further research is included in Section 5. Conclusion.

Reviewer 2 Report

The paper is well written and presents a very interesting research.

I could be improved by including the used questionnaire as an Annex, in order to better understand the statistical tables and especially how the factors were found.

The discussion and conclusion section needs more detail about the contribution to theory.

Limitations and further research needs to be included in the Conclusion section and not only in the abstract.

Author Response

Point 1: I could be improved by including the used questionnaire as an Annex, in order to better understand the statistical tables and especially how the factors were found.

Response 1: Thank you for this comment. The questionnaire items have been added to the appendix for better understanding by researchers.

Point 2: The discussion and conclusion section needs more detail about the contribution to theory.

Response 2: Thank you for this comment! After receiving this comment, we enriched the discussion and conclusion section. Please see page 8~9

Point 3: Limitations and further research needs to be included in the Conclusion section and not only in the abstract.

Response 3: Thank you for this comment! After receiving this comment, we enriched the conclusion section including limitations and further research. Please see page 8~9

Reviewer 3 Report

The paper concentrates on the important issue from the perspective of students participating in entrepreneurship club activities in college in regard to the effect of entrepreneurship on the impact of start-up intention on innovative behavior.

Strong sides of the papers:

Generally good writing skills of the author(s) - though there are sometimes problems with the style and English. Good structure of the paper. Concept of the research.

Let me reiterate author’s conclusions that “(…) to increase the sustainability of firms through start-up and to grow a desirable start-up culture, it is necessary to bring change to the awareness or perception of people about start-up and to create a good-quality entrepreneurial environment. It is important to improve the function of cultivation and metacognition through education on entrepreneurship in that it is inseparable from the cultivation of entrepreneurship from a willingness to start a business.  After all, the success of a startup is an effort for innovation.” This result indicates the path for entrepreneurship and innovation policy in the future.

However, before publication the paper needs following improvements:

The title should also indicate the country of research, which can be important for potential reader. Introduction should be extended, i.e. brief information on used methods, why this method was chosen instead of etc. Literature review and improvements in regard to literature background should be made.

Examples of articles that may be useful for the authors identifying similar contributions and better show what is the specific contribution.

Ji Young Kim, Dae Soo Choi, Chang-Soo Sung, Joo Y. Park (2018) The role of problem solving ability on innovative behavior and opportunity recognition in university students, Journal of Open Innovation: Technology, Society, and Complexity, 4(1), 4. doi: 1186/s40852-018-0085-4 Jaaffar, A.H.; Ganesan, Y.; Isa, A. (2018) Employees' Motivation to Undertake Entrepreneurship and Innovative Behavior, Global Business & Management Research: An International Journal, 10(3), pp. 782-796.

In my opinion these two references cannot be omitted.

Conclusions should be extended with the information on ideas for future research. The limitations of the research should be given in the conclusions.

To sum up, the paper has important practical value. The article’s title clearly and adequately illustrates the content. Hence, I recommend the acceptance of this article for publication after corrections.

Author Response

Point 1: The title should also indicate the country of research, which can be important for potential reader.

Response 1: Thank you for this comment! After receiving this comment, we have specifically changed the title from “The Mediating Effect of Entrepreneurship on the Impact of Start-up Intention on Innovative Behavior” to “The Mediating Effect of Entrepreneurship on the Impact of Start-up Intention on Innovative Behavior of University Students in Korea”.

Point 2: Introduction should be extended, i.e. brief information on used methods, why this method was chosen instead of etc.

Response 2: Thank you for this comment! After receiving this comment, we enriched the method section. We added Section 1.3 Research Method and Results. Please see page 4.

Point 3: Literature review and improvements in regard to literature background should be made. Examples of articles that may be useful for the authors identifying similar contributions and better show what is the specific contribution.

Ji Young Kim, Dae Soo Choi, Chang-Soo Sung, Joo Y. Park (2018) The role of problem solving ability on innovative behavior and opportunity recognition in university students, Journal of Open Innovation: Technology, Society, and Complexity, 4(1), 4. doi: 1186/s40852-018-0085-4 Jaaffar, A.H.; Ganesan, Y.; Isa, A. (2018) Employees' Motivation to Undertake Entrepreneurship and Innovative Behavior, Global Business & Management Research: An International Journal, 10(3), pp. 782-796. In my opinion these two references cannot be omitted.

Response 3: Thank you very much for this suggestion. We reworked the introduction section and added the most important relevant research references. Please see below and pages 2.

“There is a significant relationship between entrepreneurial motivation to undertake entrepreneurship and innovative behavior, and it is both intrinsic and extrinsic motivation that affects the innovative behavior [3]. In particular, people with high entrepreneurship have a positive problem solving ability in their innovation behavior [6].”

Point 4: Conclusions should be extended with the information on ideas for future research. The limitations of the research should be given in the conclusions.

Response 4: Thank you for this comment! After receiving this comment, we enriched the conclusion section including limitations and further research. Please see page 8~9

Reviewer 4 Report

This is an interesting contribution to our understanding of start-up entrepreneurial intention but I think that the authors provide insufficient rationale for their work. I am not sure, however, that they clearly describe the state of the existing literature on star-up intetion. There is a literature review, but is it clear what we already know about start-up entrepreneurial intention? The purpose of this article should be clearly defined in the introduction. The authors should explain better research subject and research context (where exactly the study was conducted) and what the selection of respondents looked like? In the limitations, it would be useful to address the question of how, context might be expected to influence the results. Does the context limit the applicability of the results so that they might be expected to be different where, for example, the respondents were from other country or from Europe (developing nation rather than an advanced one)? In the theoretical framework, is there anything in the literature suggesting that difference in context affects intention (e.g. can cultural differences be a factor? I think there is a potential for a more advanced analysis of the literature on factors influencing start-up entrepreneurship intention and  realisation. Also, something could be said about the high failure rate of start-ups (in first 3 years of performance). I also have doubts about the research sample. Almost half are people representing social sciences. It is known that startup founders in particular represent exact sciences (e.g. computer science, engineering science, materials science, etc). How to explain the selection of the sample?

Author Response

Point 1: This is an interesting contribution to our understanding of start-up entrepreneurial intention but I think that the authors provide insufficient rationale for their work. I am not sure, however, that they clearly describe the state of the existing literature on star-up intetion. There is a literature review, but is it clear what we already know about start-up entrepreneurial intention?

Response 1: Thank you for this comment. We made corrections based on the suggestion. Please see below and page 2.

“In Ajzen’s theory of planned behavior, intention is a product of interaction among attitudes toward specific behavior, subjective norms, and perceived control [15].”

Point 2: The purpose of this article should be clearly defined in the introduction. The authors should explain better research subject and research context (where exactly the study was conducted) and what the selection of respondents looked like?

Response 2: Thank you for this comment! After receiving this comment, we enriched the introduction section. Please see below and page 2.

“This study investigates whether higher start-up intention leads to innovative behavior and whether innovative behavior is increased by the medium of entrepreneurship. Also, this study tests that start-up intention affects innovative behavior and has a significant effect on the sub-factors of entrepreneurship.”

“To test our hypothesis, we conducted a survey on students of entrepreneurship club at two universities in Gyeongsangbuk-do.”

Point 3: In the limitations, it would be useful to address the question of how, context might be expected to influence the results. Does the context limit the applicability of the results so that they might be expected to be different where, for example, the respondents were from other country or from Europe (developing nation rather than an advanced one)?

Response 3: Thank you very much for this suggestion. We reworked the conclusion section including limitations and further research. Please see page 8~9

Point 4: In the theoretical framework, is there anything in the literature suggesting that difference in context affects intention (e.g. can cultural differences be a factor? I think there is a potential for a more advanced analysis of the literature on factors influencing start-up entrepreneurship intention and realisation.

Response 4: Thank you very much for this suggestion. We reworked the conclusion section including limitations and further research. Please see page 8~9

Point 5: Also, something could be said about the high failure rate of start-ups (in first 3 years of performance). I also have doubts about the research sample. Almost half are people representing social sciences. It is known that startup founders in particular represent exact sciences (e.g. computer science, engineering science, materials science, etc). How to explain the selection of the sample?

Response 5: Thank you very much for this suggestion. This study is aimed at university students who have intention to start a business in the actual idea stage of business start-up. Therefore, this study could be analysed regardless of the respondents' major direction. In future research, as the target population is expanded to the general public, it is necessary to analyse the group by dividing the group based on the individual's field of study and practical ability. These further studies have been added to the conclusion sections (See pages 12).

Round 2

Reviewer 4 Report

Dear Authors,

Thank you very much for the opportunity to read your paper focused on exploring differences in entrepreneurial start-up intentions among students in Gyeongsangbuk.

After Authors corrections, the paper is more easy to read, the proper empirical methodological approach supports the findings, and the structure of the paper is logical. After reading the paper, I hope the authors find my comments useful as they aim to improve the quality of the paper.

In my opinion, the quality of the presented manuscript still requires improvements

1. Sample and data collection –

Data collection procedure.  How the authors proceeded with the sample collection?? However, the data collection procedure is not transparent. We do not know whether the collected samples are representative for the populations, but we have no evidence that this would be the case… We would need more information concerning the transparency of the collected data.

2) Theoretical discussions

Deep contextual analysis of the literature (publications) and existing  secondary data start-up intention is  missing. The authors should carefully study the context and explain, why they chose such variables for the questionnaire? 

Author Response

Point 1: Sample and data collection - Data collection procedure.  How the authors proceeded with the sample collection?? However, the data collection procedure is not transparent. We do not know whether the collected samples are representative for the populations, but we have no evidence that this would be the case… We would need more information concerning the transparency of the collected data.

 Response 1: Thank you for this comment. We made corrections based on the suggestion. Please see below and page 4.

 2.1 Sample and data collection

“In particular, we surveyed two representative universities with the largest startups in the region, and analyzed the data collected in the field by handing out questionnaires directly to university students conducting club activities during June 2017.”

Point 2: Theoretical discussions - Deep contextual analysis of the literature (publications) and existing secondary data start-up intention is missing. The authors should carefully study the context and explain, why they chose such variables for the questionnaire?

Response 2: Thank you for this comment! After receiving this comment, we enriched the introduction section. Please see below and page 2.

2.2. Measurement of variables

2.2.1. Start-up intention

“It is worth noting that the startup intention to start a business is that all human behaviors (including business-related behaviors) take precedence and are related to the composition of the various elements, and there may be several external or internal motives [35]. This is an important task for researchers to search for factors influencing business intentions if they are one of the variables that trigger business-related behavior [36]. In addition, most of the potential entrepreneurs (ie, people who plan to carry out their business activities in the near future) are found among students. The important thing is that in order for university students to make business decisions and actions, a positive attitude and experience about business startups should be preceded [37].”
